# Species-Specific Transcription Factors Associated with Long Terminal Repeat Promoters of Endogenous Retroviruses: A Comprehensive Review

**DOI:** 10.3390/biom14030280

**Published:** 2024-02-26

**Authors:** Md Jakir Hossain, Perpetual Nyame, Kazuaki Monde

**Affiliations:** Department of Microbiology, Faculty of Life Sciences, Kumamoto University, Kumamoto 860-8556, Japan; 217r6121@st.kumamoto-u.ac.jp (M.J.H.); 214r5156@st.kumamoto-u.ac.jp (P.N.)

**Keywords:** endogenous retrovirus, transcription factor, long-terminal repeat

## Abstract

Endogenous retroviruses (ERVs) became a part of the eukaryotic genome through endogenization millions of years ago. Moreover, they have lost their innate capability of virulence or replication. Nevertheless, in eukaryotic cells, they actively engage in various activities that may be advantageous or disadvantageous to the cells. The mechanisms by which transcription is triggered and implicated in cellular processes are complex. Owing to the diversity in the expression of transcription factors (TFs) in cells and the TF-binding motifs of viruses, the comprehensibility of ERV initiation and its impact on cellular functions are unclear. Currently, several factors are known to be related to their initiation. TFs that bind to the viral long-terminal repeat (LTR) are critical initiators. This review discusses the TFs shown to actively associate with ERV stimulation across species such as humans, mice, pigs, monkeys, zebrafish, Drosophila, and yeast. A comprehensive summary of the expression of previously reported TFs may aid in identifying similarities between animal species and endogenous viruses. Moreover, an in-depth understanding of ERV expression will assist in elucidating their physiological roles in eukaryotic cell development and in clarifying their relationship with endogenous retrovirus-associated diseases.

## 1. Introduction

Retroviruses are representative examples of viruses that exist both endogenously and exogenously [1,2,3]. For example, human endogenous retrovirus-K (HERV-K) integrates its proviral DNA, which consists of four coding regions (*gag*, *pro*, *pol*, and *env*) and an accessory gene (*rec* or *np9*) flanked by long terminal repeats (LTRs), into the human genome [4,5] (Figure 1A). By incorporating their DNA sequences into the germline [5,6], they eventually become generationally inheritable endogenous retroviruses (ERVs) [7,8,9] and are subject to natural selection according to the Mendelian formula. As a result, their adverse effects are eliminated or inactivated by mutations over time [10]. In contrast, some ERVs, such as HERV-K, may still produce infectious virus-like particles [11,12,13,14,15,16]. Although ERVs remain as fossils millions of years ago, they constitute ~8.3% of the human genome [17,18,19,20,21,22].

Generally, ERVs are expressed in the placenta and embryonic tissues [23,24,25,26,27], certain cancer cells [28,29,30,31,32,33,34,35,36,37,38,39,40,41,42,43,44,45], as well as patients with schizophrenia [46,47,48,49], rheumatoid arthritis [50,51,52,53], and amyotrophic lateral sclerosis (ALS) [54,55]. Transcription factors (TFs) are the main factors involved in ERV expression in the placenta; however, multiple oncogenic signaling pathways, such as the WNT signaling pathway, are also important for ERV expression [56]. Thousands of ERVs are transcribed in somatic tissues at multiple sites. These ERV expressions in somatic tissues are leaky or driven by ERV LTRs [57]. The induction of HERV-K expression by viral infection may contribute to a variety of illnesses [58,59]. Notably, infection with Kaposi sarcoma-associated herpes virus triggers the expression of HERV-K envelope-derived proteins (Rec and NP9), resulting in viral propagation, which contributes to cellular dysfunction [60]. Furthermore, ERVs can recombine to generate solo-LTR that can induce the expression of neighboring genes [61,62] (Figure 1B). Once activated, ERVs are expressed in healthy tissues and act as potential oncogenic drivers [63,64,65]. Consequently, they function as ancillaries in processes that aid the progression of diseases, such as oncogenesis, and cellular senescence. Moreover, it may trigger diseases such as autoimmune diseases and neurological disorders [66,67,68,69,70] (Figure 1C).

ERV LTRs may serve as transcription initiation sites for their open reading frames (ORFs) through multiple cellular mechanisms [71,72,73]. Therefore, the *cis*-regulatory elements of ERV LTRs are utilized in physiological processes, including stem cell pluripotency, proliferation, and survival [27,73,74,75,76,77,78,79,80,81] (Figure 1D). Moreover, ERVs serve as host defense against exogenous retroviruses [23,82,83,84,85,86,87,88,89,90,91,92,93] (Figure 1E–H). Presumably, a wide range of TF-binding motifs routinely regulate LTR activation, which is required for triggering HERVs, murine endogenous retroviruses (MERVs), porcine endogenous retroviruses (PERVs), simian endogenous retroviruses (SERVs), zebrafish endogenous retroviruses (ZFERVs), and yeast retrotransposons (Ty1/3) [94,95].

As demonstrated in earlier studies, TFs, or sequence-specific DNA-binding factors, are molecules that bind to a specific DNA sequence and modulate the transcription of DNA into mRNA [96,97]. In multicellular organisms, TFs are pivotal for cellular differentiation, and disease progression [98,99]. TFs possess species-specific DNA-binding domains, which facilitate their binding to target sequences [100]. Therefore, research on the uniqueness of TFs to LTRs, the variety of LTRs in several species, and the positioning of LTRs in the genome are indispensable for unraveling the mystery of ERVs. Currently, ERV LTR-activated TFs and LTR-driven genes have been numerously reported based on the reporter assay in vitro, the next-generation sequencing analysis, and the bioinformatic analysis as described below (Figure 2A). However, a great deal of information that has been exhaustively analyzed must be organized for new discoveries in future. This review highlights impactful data depicting different ERV LTRs activation by TFs as well as associated cellular dysfunctions across different species. We hope that this review will serve as a comprehensive resource for the practical application of regenerative medicine using induced pluripotent stem cells (iPSCs) and xenotransplantation using porcine tissue, as well as for the development of cancer and aging research (Figure 2B). We also hope that the coexistence of the host with the virus will provide clues to elucidate how the virus has contributed to biological evolution.

## 2. HERV LTRs Are Required for the Transcription of Neighboring Genes

LTRs are essential for transcription in multicellular organisms [71,72,73]. Based on the LTR sequence, HERV-K (HML-2) LTR is classified into LTR5A, LTR5B, and LTR5Hs subtypes [101]. Indeed, the HERV-K core promoter elements (U3, R, and U5) are essential to make the promoter functional [102,103,104]. As every provirus has undergone distinct mutations over the course of millions of years, these LTR sequence modifications may affect the binding of TFs to LTR-binding sites, which in turn may lead to the distinctive expression of HML-2 [105]. Chromatin state analysis has revealed that different HERV LTRs enriched in the enhancer regions are distinct across cell types [106]. The expression of neighboring genes from ERV LTR was analyzed by the activation/silencing of the HERV-K LTR5Hs using CARGO with CRISPR activation (CRISPRa) or interference (CRISPRi) [62]. The results indicated that the chromatin state of its neighboring genes is remarkably changed and that LTR5Hs acts as distal enhancer or suppressor to regulate the expression of at least 275 genes [62]. Furthermore, the expression of neighboring genes, which are located either proximal or distal within a range of ~200 kb from LTR5Hs, is regulated by LTR5Hs activation or silencing [62]. In summary, because LTRs affect the expression of neighboring genes within a 200 kb range, identifying the elements involved in transcription from the LTRs may allow us to determine causal relationships with various ERV-related diseases.

### 2.1. TFs Associated with LTR Activation of HERV-K

p53 directly upregulated the promoter activity of LTR5Hs rather than LTR5A and LTR5B in cervical carcinoma (HeLa) and HEK293T cells [101]. Luciferase reporter and ChIP assay results proved that the two binding sites for p53 on LTR5Hs are important for upregulating transcription and activating other LTRs [101]. Although p53 plays an important role in suppressing cancer progression, the mechanism by which it activates HERV-K remains unclear. A recent study reported that the presence of anti-HERV-K Env antibodies is important for lung cancer immunotherapy [45]. Thus, if p53-mediated HERV-K activation is involved in lung cancer immunotherapy, this will be a topic for future research.

HERV-K LTR5Hs (HERV-K113) has a full-length ORF and is located on chromosome 19p13.11 [14]. The prevalence of HERV-K113 mRNA is higher (29%) than that of other HERVs in various cancer cells, particularly teratocarcinoma cells [14]. Specific TFs are necessary to activate this LTR-driven genome, which include Sox2, Oct4, and Nanog (Table 1) [23,107,108]. However, the individual expression of Oct4 and Nanog is insufficient to activate HERV-K LTR5Hs [107], while the combined expression of Oct4 and Nanog, along with Sox2, markedly enhances the transactivation capability [107]. Historically, Sox2 has been shown to control cancer stem cell maintenance and self-renewal, fostering oncogenic signaling [109,110,111]. Interestingly, HERV-K expression is significantly upregulated in germ cell tumors, melanomas, and ovarian cancers compared to that in healthy tissues [42,112,113,114]. These findings suggest that the transactivation of HERV-K LTR5Hs by Sox2 is involved in numerous malignant tumors.

Double homeobox 4 (DUX4) is another prominent TF that activates HERV-L LTRs in rhabdomyosarcoma cells (muscle cancer cells) [115,140,141,142]. Indeed, ERV transcripts were detected in the skeletal muscles of individuals diagnosed with facioscapulohumeral muscular dystrophy (FSHD) [115,142]. RNA-seq and ChIP assay results suggest that DUX4 overexpression causes the initial manifestation of FSHD [115]. Notably, DUX4 induces HERV-K LTR5Hs on chromosome 7p22 in glioblastoma and myoblastoma cell lines (Table 1), which may result in neurological diseases [17,115].

The resurrection of HERV-K LTRs is alarming for genomic stability and may be associated with the initiation and upregulation of renal cell carcinoma (RCC) [116,143,144,145]. Numerous studies have reported the expression of different HERVs in patients with RCC [146]. In patients with RCC, HERV-K LTR5Hs LTR is located on the long arm of chromosome 6, and HERV-E LTR is significantly expressed in the presence of hypoxia-inducible transcription factor (HIF), which binds to transcriptionally active LTR elements [146]. Thus, evidence suggests that HIF-dependent reactivation of dormant promoters embedded within endogenous retroviral LTRs is a potential contributing factor to dysregulated gene expression in RCC [116].

Previous research on the binding motifs of TFs revealed the presence of microphthalmia-associated transcription factor-M (MITF-M) on HERV-K LTR as well as some other retroviral LTRs [123]. The expression of HERV-K on chromosome 1 was shown to be significantly higher in the presence of MITF-M in melanoma and HEK293 cells [123,147]. Further analysis uncovered that the sequence of MITF-M (MITF-1, MITF-2, and MITF-3), TATA, and Inr is quite conserved (5′CACATG3′) in over a hundred HERV-K LTRs [123]. Mutants of the MITF-1, -2, or -3 motifs at the HERV-K LTR were unable to initiate transcripts in malignant melanoma cell lines (MeWo cells), which express large amounts of endogenous MITF-A and -M [123]. MITF-M binds to the LTR at the 5′ and 3′ ends, activating both HERV-K LTRs [123,147,148]. This suggests that MITF-M alters the expression of neighboring genes in malignant melanoma cells.

HERV-K env RNA expression is upregulated in patients with atypical teratoid rhabdoid tumor (AT/RT) by deletion or mutation of integrase interactor 1 (SMARCB1) [95]. Based on computational assessment using the PROMO software v8.3 of TRANSFAC, c-Myc protein binding sites were identified within the HML-2 LTR5Hs sequences [149]. SMARCB1 is a transcriptional repressor of HIV-1 LTR [150]. The active LTRs in loci 7p22.1a and 7p22.1b are highly expressed in patients with AT/RT as a result of c-Myc binding to the HERV-K LTR (Table 1) [95]. Similar results were found in 293T cells [150], suggesting that SMARCB1 is a repressor and c-Myc is an activator for HERV-K LTR in AT/RT.

HERV-K is actively involved in breast cancer progression in the presence of female sex hormones estradiol and progesterone [118]. Although estradiol and progesterone have other biological functions, they synergistically activate HERV-K through their receptors in T47D human breast cancer cells [118]. Electrophoretic mobility shift assay (EMSA) and co-immunoprecipitation assay showed that the progesterone receptor (isoform B) binds to the progesterone response element within LTR5Hs [118]. Conversely, overexpression of Oct4 significantly (two-fold) enhanced HERV-K10 transcription, and progesterone treatment synergistically activated HERV-K10 LTR in primary mammary epithelial cells [118]. Thus, it can be assumed that HERV-K, which is activated by female sex hormones, drives breast cancer progression [118].

The TF Yin Yang 1 (YY1) is highly conserved from African clawed frogs (*Xenopus laevis*) to humans [151,152]. It is ubiquitously expressed in pluripotent differentiated cells, mouse teratocarcinoma cells (F9), mouse fibroblasts (NIH3T3), rat embryo fibroblasts, and HeLa cells [153,154]. YY1, together with different cofactors, mediates the activation, repression, or initiation of transcription in ERVs [155]. YY1 binds to a motif within 62nt-83nt of the HERV-K LTR5Hs and acts as an enhancer-binding protein in human teratocarcinoma (GH and Tera2), hepatocarcinoma (HepG2), and HeLa cells [120]. Additionally, YY1 enhancer complexes activate HERV-K LTR5Hs by binding at the 5′ end of the U3 region [120]. In contrast, YY1 induces silencing of exogenous and endogenous retroviruses by recruiting tripartite motif-containing protein 28 (TRIM28) and its complex in mouse embryonic cells [121,156,157,158,159]. Thus, ERV LTR is activated or suppressed depending on the transcriptional activator or repressor that binds to YY1; therefore, it is important to investigate the relationship between YY1-binding cofactors and diseases.

HERV-K is also associated with numerous neural diseases [55]. HERV-K env leads to neuronal injury in the presence of the transactive response DNA-binding protein 43 (TDP-43) [119]. The co-expression of HERV-K RT and TDP-43 proteins has been observed in most neurons, with a significant positive correlation between them [160]. Five different TDP-43 binding sites were found on the HERV-K LTR, which regulate its activation [119]. Human neural cells transfected with TDP-43 showed enhanced HERV-K expression, whereas knockdown of endogenous TDP-43 resulted in decreased HERV-K expression [119]. In addition, the levels of antibody against HERV-K were markedly increased in patients with ALS, multiple sclerosis, and Alzheimer’s disease [161]. Moreover, specific post-translational modifications of TDP-43 may affect HERV-K expression patterns. For example, formation of TDP-43 aggregates alters HERV-K RT expression and cellular localization of viral proteins [117]. Misfolded TDP-43 is aggregated and transmitted in patients with ALS [162]. Interestingly, prion-like proteins (e.g., yeast Sup35 prion NM domain and Tau microtubule-binding domain) are transmitted by ERVs such as HERV-K and HERV-W [70]. Thus, if ERVs form particles and become active in other cells by transmitting TFs listed in Table 1, including TDP-43, they may be involved in diseases such as ALS.

HERV-K LTRs and HERV-E.PTN are TATA-independent promoters regulated by three GC boxes that serve as binding sites for Sp1 and Sp3 [103,163]. Sp1/Sp3 are TFs for TATA-less promoters that are available early after zygote formation [164]. HERV-K LTR activation was reduced by approximately 20% and 50% following knockdown of Sp1 and Sp3, respectively, in human melanoma (Mel-C9) and teratocarcinoma (GH) cell lines [103]. Mutation of the Sp1-binding motif (451–462 nt) in HERV-E.PTN markedly reduced transcriptional activity in choriocarcinoma cells of the fetal placenta (JEG-3, BeWo, and JAR). Since HERV-W and HERV-FRD Env play critical roles in placental development [130], Sp1 and Sp3 may trigger HERV-W transcription in placental cells. However, the active HERV-W LTR has a mutation in the Sp1-binding motif, while the inactive HERV-W LTR has an intact Sp1-binding motif in lung fibroblasts (LC5) (Table 1) [129]. Thus, transcriptional activation by Sp1 and Sp3 is complex, and their regulatory mechanisms may differ depending on other transcription elements and cell lines.

### 2.2. Viral Infection Activates HERV-K and HERV-W Expression

The HTLV-1 Tax protein, detected in patients with myelopathy or tropical spastic paraparesis (HAM/TSP), is a powerful trans-activator capable of inducing many cellular genes through its activation domains [165]. HTLV-1 Tax trans-activator could activate the LTRs of different HERV families (HERV-W8, HERV-H, HERV-K, and HERV-E) in T cell lines [125]. Indeed, various Tax mutants have been shown to affect LTR activation to different degrees. Moreover, modulation of these LTRs by Tax activated CREB and possibly NF-κB in Jurkat cells, which in turn may positively regulate the transcription of HERV genes or other proximal cellular genes in HTLV-1-infected patients [125].

Several TFs listed in Table 1, such as AP-1, CREB, CEBP (C/EBP_α_), c-Rel, NF-AT, CEBP_β,_ NF-κB (p50:p52), Rel-A, p53, YY1, c-Myc, Sp1, Sp3, and signal transducer and activator of transcription 1 (STAT1), potentially interact with HERV-K LTRs. Interestingly, multiple NF-κB sites are present in the HERV-K promoter [166,167]. Luciferase reporter gene and ChIP assay analyses showed that NF-κB mediates the activation of HERV-K via HIV-1 Tat protein [122]. As the HERV-K LTR contains potential NF-AT binding sites, NF-AT activation could additionally contribute to HERV-K Tat-driven expression and might compensate for the absence of NF-κB activity. These reports suggest that both NF-κB and NF-AT activation in response to HIV-1 Tat drives transcription from the HERV-K promoter in HIV-1-infected patients [122]. HIV-1 release and infectivity are reduced by coassembly between HIV-1 Gag and HERV-K Gag in the HERV-K Gag-overexpressing cells [83,84] (Figure 1G). The endogenous retroviruses might be activated by the infection of exogenous retroviruses to protect the host cells from exogenous retroviral threats.

LTR-driven HERV-W transcription is activated by herpes simplex virus type 1 (HSV-1) immediate early protein (IE1) through the cellular TF Oct-1 [128]. In addition, anti-HERV-W antibody level is elevated with anti-Epstein–Barr virus antibody in the patients with autoimmune demyelinating disorders [168]. Two Oct-1-binding motifs are conserved in the HERV-W LTR series [129]. Specifically, few HERV-W LTRs are stimulated in the presence of IE1, which potentially upregulates the expression of other genes [128]. Furthermore, HSV-1 immediate early protein (ICP0) increases HERV-K transcription via the AP-1-binding motif in the LTR [169]. Therefore, the relationship between various HSV-1-associated diseases and HERVs is a topic for future research.

Subsequent research has clarified that HERV-W LTR is also activated by influenza A/WSN/33 infection [170]. Although its underlying mechanism remains to be elucidated, the induction of HERV-W was found to depend on the cell line, because the expression of HERV-W *gag* and *env* genes was relatively enhanced in influenza-infected CCF-STTG1 and U937 cells, but not in 293F cells. Notably, interferon beta (IFN-β) level positively correlated with HERV-W expression in the infected cells [170]. This suggests that IFN-induced HERVs expression cannot be disregarded.

### 2.3. Interferon-α, and γ Trigger HERV-K Expression

Superantigen (SAg) IDDMK_1,2_ 22, associated with type-1 diabetes, is derived from HERV-K18 mapped on chromosome 1q21.2-q22 [171]. IFN-α treatment upregulated the expression of SAg and HERV-K18 in T cells, while a cocktail treatment of IFN-α and IFN-γ markedly increased HERV-K18 expression [126], suggesting that HERV-K18 is induced by IFN-α and this induction can be amplified by IFN-γ priming.

IFN-γ signaling-induced HERV-K102 expression has been demonstrated in various cell lines [124]. Expression of HERV-K genes was higher in patients with cutaneous leishmaniasis, and IFN-γ level is known be elevated in these due to leishmania parasite infection [124,172]. Transposase-accessible chromatin sequencing (ATAC-seq) and ChIP sequencing analyses showed the HERV-K102 expression was upregulated via solo-LTR LTR12F upon treatment with IFN-γ in HeLa cells [124]. IFN regulatory factor 1 (IRF1) and, potentially, STAT1 are conjugally recruited to activate LTR12F located upstream of HERV-K102 [124]. Upon activation of LTR12F along with the enhancer histone H3 dimethylation of lysine 4 (H3K4me2), HERV-K102 activates following IFN-γ signaling [124]. This suggests that HERV-K102 activation is sensitive to IFN-γ and is regulated through IRF1 recruitment followed by LTR12F activation. The notable finding is that HERV-K102 expression is upregulated by utilizing upstream solo-LTR rather than its own LTRs, and that IFN-γ and IRF1 recruitment are the trigger for solo-LTR activation.

## 3. TFs Associated with LTR Activation of Other HERVs

HERVs are classified according to the type of tRNA that binds to the primer binding site (PBS) located downstream of the 5′LTR. For example, in the case of HERV-K, HERV-K utilizes a lysine (K) tRNA, and in the case of HERV-W, it is tryptophan (W) tRNA [173].

### 3.1. HERV-E Is Activated by HIFs, Nuclear Factor of Activated T-Cells 1 (NFAT1), and Estrogen Receptor Alpha (ER-α)

HERV-E (CT-RCC-8 and CT-RCC-9), located on the long arm of chromosome 6 (GenBank accession number AL133408), is selectively expressed in most RCCs [174]. Notably, HERV-E Env protein has antigenic properties that immunologically provoke cytotoxic T cells to kill RCC cells both in vitro and in vivo [144,174]. HERV-E provirus was shown to be resurrected in the clear cell subtype of RCC (ccRCC) upon inactivation of the von Hippel–Lindau (*VHL*) gene, which is a tumor suppressor gene [131]. Furthermore, this activation of HERV-E can be stabilized by HIF-2α but not HIF-1α [131]. Computational analysis showed that the binding motif of HIF-2α is located on the HERV-E LTR, and in vitro investigation found a direct correlation between the expression levels of HIF-2α and HERV-E in ccRCC [131]. Furthermore, ChIP analysis revealed a direct binding association between HIF-2α and HERV-E 5′LTR [131]. Taken together, these findings suggest that of *VHL* suppression-activated HERV-E, which could be promoted and stabilized by HIF-2α in RCC [131].

Elevated ERV protein levels have also been found in patients with systemic lupus erythematosus (SLE) [175], and autoreactive CD4^+^ T cells play a principal role in this disease [176]. The TFs NFAT1 and ER-*α* bind to the HERV-E clone 4-1 LTRs located on chromosome 19p12 [132]. Overexpression of NFAT1, and ER-*α* activated the HERV-E clone 4-1 5′LTRs in CD4^+^ T cells of patients with SLE, as revealed by luciferase reporter and the ChIP assay analyses [132]. In contrast, the antisense RNA miR-302d transcribed from the 3′LTR of HERV-E clone 4-1 induces DNA hypomethylation and is associated with SLE [132]. The hypomethylation activity of HERV-E has been confirmed in patients with SLE by COBRA validation [177,178]. Moreover, studies have reported that HERV-E mRNA, but not HERV-K and HERV-W, is increased in CD4^+^T cells from patients with SLE, suggesting a crucial role for HERV-E in the development of SLE [179].

### 3.2. HERV-L Is Activated by the Hepatocyte Nuclear Factor (HNF-1)

Solo-LTR (MLT2Bs), a member of the HERV-L family, is a promoter of the human beta-1,3-galactosyltransferase 5 (*β3Gal-T5*) gene, which is involved in type 1 Lewis antigen synthesis [79]. The ERV-L LTR promoter is most active in the gastrointestinal tract and mammary glands [79]. HNF-1 binds to ERV-L LTR and acts as a TF [79]. Two predicted sites for HNF-1 binding were identified at nucleotide positions 7–21 and 33–46 using the TRANSFAC TF database [79]. HNF-1 is expressed in tissues where the LTR promoter is active, including the intestine, stomach, kidneys, liver, and thymus [180]. LTR-driven β3Gal-T5 is expressed in the mammary glands, small intestine, trachea, colon, thymus, stomach, kidneys, liver, and lungs [79]. Of the two HNF-1-binding sites in the HERV-L LTR, the second site (position 33–46) was more important for the specific activation of the LTR promoter in a colorectal cancer cell line (LoVo). HNF-1, therefore, represents a candidate TF responsible for the tissue-specific activation of the HERV-L LTR promoter in various cancer cells, such as colorectal cancer cells.

### 3.3. HERV-W Is Activated by Glial Cells Missing-a

Human glial cells missing-a/1 (GCM-a/1) and murine glial cells missing-a (mGCM-a) are placenta-specific TFs required for placental development [181,182]. The HERV-W env syncytin-1, positioned on chr7q21-chr7q22, is regulated by GCM-a in BeWo and JEG3 cells [183]. Binding motif analysis identified two GCM-a-binding sites (25538-25545, 28026-28033) upstream of the HERV-W 5′LTR (GenBank accession no. AC000064; 7q21-7q22). Therefore, GCM-a can easily transactivate syncytin-1 gene, especially in trophoblasts [130]. The close proximity between the GCM-a-binding site and the LTR ensures the formation of an integral syncytiotrophoblast layer in the placenta [130].

Notably, GCM1 expression is activated via the WNT signaling pathway [184], suggesting that GCM1 and WNT signaling upregulate HERV-W expression in placental cells. In contrast, the HERV-W env protein increased the proliferation and viability of immortalized human uroepithelial cells. Results of colony-formation experiments and in vivo tumor xenografts suggest that syncytin-1 overexpression, due to two mutations in the 3′-LTR (T142C and A277G), has oncogenic potential in the urothelial cell carcinoma (UCC) [127]. The T142C mutation favors the binding of TF c-Myb to HERV-W 3′-LTRs and upregulates syncytin-1 overexpression [127]. This suggests that further research on HERV-W is important not only to elucidate the biological evolution due to placentation, but also to explore its relationship with development of cancers, such as UCC.

### 3.4. HERV-H Is Activated by Sox2, Nanog, and Oct4

HERV-H is present in the human genome as 100 full-length copies and >1000 solo-LTR copies. The solo-LTR of HERV-H was integrated into a gasdermin-like protein (GSDML) located on chromosome 17q21 during hominoid evolution. This solo-LTR drives GSDML–GSDM gene transcription in human gastric and breast cancers [133,134]. The expression and promoter activity of HERV-H LTR depend on the cell type [129]. Moreover, the HERV-H LTR has different TF-binding sites, such as Sp1, GC box, and TATA box [135,136]. RNA-seq revealed a binding association between HERV-H and the TFs Sox2, Nanog, and Oct4. During differentiation of embryonic stem cells (ESC), this binding association was greater for Sox2 than that for Nanog or Oct4 [185]. Ectopic expression of LBP9, Oct4, Nanog, and Klf4 activated HERV-H transcription in human primary fibroblasts, whereas overexpression of Myc or Sox2 failed to activate HERV-H [137]. Disruption of HERV-H transcripts compromises self-renewal, suggesting an important role for HERV-H expression in pluripotency [137]. Interestingly, the HERV-H transcripts (ESRG) play a crucial role in the maintenance of human pluripotency in iPSCs [186]. In summary, further studies on the transcriptional regulation of HERV-H and the function of its transcripts are important for the development of regenerative medicine using iPSCs.

### 3.5. HERV-S71 (HERV-T) Is Activated by GATA4 and FOXA2

GATA4 and Forkhead box protein A2 (FOXA2) are two distinct TFs that drive the regulatory network of the endoderm [187,188]. Both TFs were upregulated during definitive endoderm (DE) differentiation of hESCs. LTR6B, a DE-specific enhancer of ERVs, contains GATA4- and FOXA2-binding motifs and flanks HERV-S71-int [138], which is classified as HERV-T [189]. GATA4 and FOXA2, which bind to LTR6B, activated neighboring genes located in the vicinity of ~50 kb in DE cells, as determined by ChIP seq. This was supported by the finding that FOXA2 depletion downregulated the expression of LTR6B in DE cells, as shown by ATAC-seq and H2K27ac ChIP-seq analyses [138]. Thus, GATA4 and FOXA2 play a critical role in activating neighboring genes by binding to ERV LTR6B [138].

### 3.6. Solo-LTR (ERV-9) Is Activated by GATA-2, NF-Y, and MZF1

The ERV-9 solo-LTR is present upstream of the DNase I-hypersensitive site 5 (HS5) in the human β-globin locus control region and acts as an elite enhancer of cis-linked genes in the oocytes and progenitor stem cells [190]. EMSA revealed the binding motifs of GATA-2, ubiquitous NF-Y, and hematopoietic MZF1 within the ERV-9 solo-LTR in the erythroid cell line K562 [139]. NF-Y binds to the CCAAT motif on the ERV-9 LTR and recruits MZF1 and GATA-2 to form a complex [139]. This complex stabilizes their binding to neighboring GTGGGGA and GATA motifs. Subsequently, NF-Y binds to the complex, assembling an efficient LTR enhancer complex, which can accelerate the transcription of the β-globin gene [139].

## 4. TFs Associated with ERV LTR Activation in Other Species

As described HERV-K above, MERV-L LTR is also required for the transcription of two-cell embryo genes in mice [191]. Interestingly, the knockdown of MERV-L expression reduces the inner cell mass (ICM) genes (Oct4, Sox2, and Nanog) and trophectoderm differentiation genes (Tead4, Tcfap2c, and Cdx2) [192]. These findings suggest that ERVs, what are expressed during embryo, might be key factors for preimplantation development. On the other hand, the interference between ERV and exogenous retroviruses has also been highlighted in mice and cats. For instance, Friend-virus-susceptibility-1 (*Fv-1*), which is located on mouse chromosome 4 [193], restricts the murine leukemia virus (MLV) replication in mouse embryo cells [90,91] and determines the tropisms (MLV-N or MLV-B) [194] (Figure 1F). The *Fv-1* sequence has 60% identity with HERV-L *gag* gene [195]. The host defense of ERVs might have persisted through the process of biological evolution.

### 4.1. TFs Associated with LTR Activation of MERVs

Histone and DNA methylation result in the transcriptional suppression of MERVs during the initial phase of embryogenesis [196,197,198,199,200,201,202,203,204]. TRIM28, also called KRAB-associated protein 1 (KAP1), reportedly restricts gene expression via synergy between NuRD histone deacetylase complex, heterochromatin protein 1 (HP1), and histone methyltransferase SETDB1 [205]; however, to date, only a handful of its regions have been identified [206]. Nevertheless, it has been determined that the inhibitory influence of KAP1 on ERV expression can be manifested in three different ways. One way is that KRAB-zinc finger protein (ZFP809) binds to PBS of MERV and recruits the TRIM28 [207,208]. The TRIM28 recruitment is key for the silencing of endogenous retroviruses in mouse ESCs [208,209]. The other way is that KRAB-zinc finger protein (ZFP708) also binds to TRIM28 and plays a key role for the embryonic development via the ERV-K elements (RMER19B) silencing [210]. Last, YY1 binds to LTR and recruits the TRIM28 as described above [121,156,157,158,159]. On the other hand, TRIM28 can not only inactivate a gene during the initial events of embryogenesis, but also erase the murine leukemia virus in embryonic cells [207,208,211,212]. The loss of KAP1 causes remarkable overexpression of several types of MERVs (e.g., the IAP element) and neighbor genes from MERVs in mouse embryonic stem cells (ESCs) and initial events during embryogenesis [213,214,215].

MERV-L expression is high in embryonic stem cells [191]. The TF Zscan4c, aided by its zinc finger domains, acts as an inducer of the embryonic genes in 2-cell/4-cell embryos as well as MERV-L LTR (MT2) [216]. In addition to acting as a trigger for MT2, Zscan4c drives cellular regulation by stimulating the neighboring genes of MT2 (2C genes) in two-cell/four-cell embryos [216]. Zscan4c binds to the GLTSCR1/like-containing BAF complex (GBAF), a chromatin remodeling complex, via its SCAN domains to efficiently initiate MT2 functions, as determined by ChIP analysis [216]. On the other hand, MERV-L LTR is activated by DUX in the 2C-embryo-like cells [140,141,217,218,219]. In addition, p53 is required for the DUX expression in FSHD cells [220]. The DUX expression is repressed by the recruitment of Nucleolin/Kap1 with LINE1 RNA, thus the LINE1-Nucleolin complex indirectly represses the MERV-L LTR [221]. Moreover, PIM3 represses MERV-L via the hypo-phosphorylated HDAC4/5 [222]. ERVs may also exhibit a noticeable diversity in their expression models. In particular, these differences in ERV expression are observed at the different stages of embryogenesis, and the expression patterns are diverse among species. A vivid illustration of this can be seen in humans, where LTR14B, an affiliate of the HERV-K family, and HERV-L (MLT2A1) are augmented during the two-cell phase and in the four-cell/eight-cell embryo, respectively [216,223,224].

MERV-L LTRs are key for the expression of several cellular genes during zygotic genome activation (ZGA) and blastocyte-phase gene silencing [225]. Mutations in a lysine-specific demethylase (LSD1/KDM1A) boost the expression of ZGA-associated LTR, leading to an increase in cell propagation capabilities [225]. In ESCs, LSD1/KDM1A regulates the addition of methyl and acetyl groups during histone methylation of LTR sequences, resulting in MERV silencing. In contrast, MERV-L expression is high in ES and blastocyst-stage embryonic cells deficient in LSD1/KDM1A [225].

MERV-L expression is reported to largely depend on reduced expression-1 (Rex1), also known as Zfp42. Rex1/Zfp42 in murine ES cells attenuates MERV-L expression [226]. This mechanism of MERV-L deregulation is a constituent of the phenotypic primordial aberrations observed in Rex1/Zfp42-depleted mouse ESCs [226]. Rex1/Zfp42 is expressed in a wide variety of cells; it is expressed in pluripotent cells (predominantly in undifferentiated ES cells), multipotent adult progenitor cells, amniotic fluid (of testis germ cells, ICM), and derivatives of the trophectoderm from mouse embryos [227,228,229]. Moreover, Rex1/Zfp42 monitors MERV-L expression at the preimplantation developmental stage. Numerous regions responsible for TF binding have been identified in murine LTRs (e.g., RLTR9B2, RLTR9D, and RLTR9E) [230]. In particular, Esrrb, Klf4, and Sox2 enhance the expression of RLTR9B2, RLTR9D, and RLTR9E, respectively [230]. Esrrb, Klf4, and Sox2 are well-known reprogramming factors and master regulators of pluripotency (Table 2). In summary, the regulation of MERV-L by repression factors (Rex1/Zfp42 and LSD1/KDM1A) and activation factors (Esrrb, Klf4, and Sox2) may be crucial for reprogramming, stable pluripotency, and/or blastocyst development.

Studies on trophoblast stem cells (TSCs) from rats and mice have added to the understanding of the molecular mechanisms underlying placental development [231]. RLTR13D5, a member of the ERV family, generates hundreds of enhancers in mice that communicate with Elf5-core proteins, Eomes, and Cdx2, serving as the foundation of the TSC regulatory complex. Histone H3 lysine 4 monomethylation (H3K4me1) and histone H3 lysine 27 acetylation (H3K27ac) are two examples of these mediators [231]. In addition to RLTR13D5, exceedingly high levels of Eomes, Elf5, and Cdx2 were detected in RLTR13B4. These findings suggest that RLTR13D5 may coordinate gene expression in the rat placenta in the presence of Eomes, Elf5, and Cdx2 [231].

### 4.2. TFs Associated with LTR Activation of PERVs

PERVs have been inserted into the genome of pigs [249]. Owing to the presence of human PERV-A receptors 1 and 2 (huPAR1 and huPAR1, respectively) [250], human cells are susceptible to many types of PERVs, such as PERV-A, PERV-B, and PERV-A/C [249,251,252,253]. Therefore, xenotransplant recipients from pig donors with PERV expression cannot ignore the risk of PERV infection [232]. PERVs adapted to human cells do not produce mutations in *env* but alter the length of the LTRs [232]. For example, nuclear factor Y (NF-Y)-binding motif (CCAAT box) has been identified within PERV LTRs in 293T cells [232,254,255]. Similarly, LTR3, a member of the PERV-A family, encodes the transcription activators Nkx2-2 and Elk-1 [256]. According to the TRANSFAC database, several TFS (Sox5, Ets-1, Evi1, GATA, v-Myb, and CEBP; Table 2) have been identified within the PERV-C LTR [233]. Although studies have investigated the mechanism of transcriptional activation of PERVs after adaptation to human cells, the pathogenicity of PERVs that are activated in human cells remains unclear.

### 4.3. TFs Associated with LTR Activation of SERVs

SERVs exist as two types: Cer-SERV-1 and Cer-SERV-2 [235]. The sequences within the LTR of Cer-SERV-1 are approximately 484 nt in length and encode five to eight TF-binding motifs (CREB, CDP, E2F, AREB6, FoxD3, FoxJ2, and Brn-2), shown in Table 2 [235]. As FoxD3 and FoxJ2 are expressed in embryonic stem cells and during early embryonic development, respectively [234,257], Cer-SERV-1 may also be expressed in embryonic stem cell and during early embryonic development. Cer-SERV-2 encodes other TFs (AREB6, COMP1, CREB, NF-1, RFX1, Pax6, and v-Myb) than those encoded by Cer-SERV-1 [235]. Therefore, the transcription of Cer-SERV-2 may be regulated by other elements in different cells. The monkey-specific LTR (MacERV6-LTR1a) is briefly activated in blastocysts before implantation and is silenced after implantation. MacERV6-LTR1a recruits Esrrb, but not Klf4, Oct4, Sox2, SMAD3, or HNF4A, to activate transcription [236]. Importantly, similar to other ERVs, SERVs are expressed early during development.

### 4.4. TFs Associated with LTR Activation of Bovine ERVs (BoERVs)

Genomic expression of transposable elements (TEs) depends largely on their translocation [258]. The LTR regulatory element of the TEs plays a vital role in this process [4]. Genomic studies have shown that 27% of the bovine genome consists of TEs [259]. These TEs contribute to the genomic structure and evolution of cattle [259]. The insertion of TEs may result in neighboring gene modulation and chromosomal duplications [237]. However, the expression of BoERVs varies according to developmental stage, organ, and tissue type [237]. For example, BoERVs located on chromosomes 2 and 4 are highly expressed in endocrine glands [237]. Gene ontology analysis revealed that the LTRs of BoERV16, BoERV3, and BoERV9 contain a binding motif for the TF retinoid-X receptor alpha (Rxra), which upregulates BoERV expression in the thyroid [237]. In contrast, the ruminant-specific TEs MER41_BT and Bov-A2 are mainly expressed in bovine eight-cell embryos during development [240]. ATAC-seq analysis revealed that pluripotency factors/TFs, namely Oct4, NFY, Klf4, OTX2, TEAD, and STAT1, bind to MER41_BT and STAT1 [238], while POLR2A binds to Bov-A2 [239], thereby upregulating the expression of BoERVs. The basic principle of retroviral replication is binding of TFs to the U3 region of retroviral LTR promoter [260]. Bovine retrovirus BoRV CH15 is thought to be endogenous and causes neural disease upon activation. In silico analysis revealed the presence of binding sites for a putative TF nuclear factor-1 (NF-1) in its LTRs; NF-1 likely activates BoRV CH15 expression, damaging the brain neurons and causing cattle encephalitis [240]. Thus, further studies on TFs might be key to understanding its incentive as well as maintaining bovine health.

### 4.5. TFs Associated with LTR Activation of Feline Endogenous Leukemia Virus (enFeRVs)

Endogenous feline leukemia virus (enFeLV), which is dispersed throughout the cat genome [261], has a full-length pro-viral genome [262]. The enFeLV transcript is expressed in almost all cats [263]. Active enFeLV accelerates the diseases progression, for instance, autoimmune disease [264]. Therefore, the activation of enFeLV is concerning in domestic animal health, especially in cats. The viral load of enFeLV is elevated in the FeLV-infected cats [265]. ERV-DCs, which is the latest characterized class of enFeLVs [266], express a couple of truncated env proteins known as Refrex-1 [93]. Refrex-1 has a strong inhibitory effect on the activation or reemergence of the ERV-DCs [93] (Figure 1E). In addition, enFeLV microRNA is highly expressed in PBMCs and may inhibit the exogenous FeLV replication in PBMCs [92]. On the other hand, the transcription factors responsible for the LTR activation of enFeLV are not well revealed yet. The higher CpG methylation in the ERV-DC10 LTR reduced the expression of ERV-DC significantly [266], hence, CpG methylation controls the LTR activity of feline ERVs. The ERV-DC LTRs have been classified based on the differences in nucleotide sequences [266], and these nucleotide substitutions might influence the basal promoter activity of LTRs [266]. Further study on transcription faction is essential to unveil the physiological significances of interference between enFeLV and exogenous FeLV.

### 4.6. TFs Associated with LTR Activation of ZFERVs

Expressed-Zebrafish-Retroelement group 1 (EZR1) comprises approximately 8% of the zebrafish genome [267]. EZR1 potentially encodes LTR, *integrase,* and *env* genes but not structural genes such as *gag* [241]. EZR1 transcripts are detected in several adult tissues, such as the heart, and whole embryos [241]. Several TFs (TCF-11, Nkx-2.5, GATA-1, MZF-1, Ikaros-2) (Table 2) are candidate elements for EZR1 transcription; in particular, the zebrafish Rel family protein homologs interact with a putative NF-κB-binding motif within the LTRs [241]. Since EZR1 is expressed in a wide range of cell types via many potential TFs, the silencing mechanisms of EZR1, such as Ziwi [267], may be important for understanding the evolution of organisms.

### 4.7. TFs Associated with LTR Activation of Yeast Retrotransposons

In *Saccharomyces cerevisiae*, yeast retrotransposons Ty1–Ty5 have been discovered, with Ty1/Copia being the most abundant [268]. Ty1 elements contain two ORFs: TYA (*gag*) and TYB (*pol*) [269]. Forty years ago, Ty1 was discovered as an LTR-retrotransposon that transposes through RNA to move its genes [270]. Subsequently, several LTR-retrotransposons that transpose via RNA have been discovered [107,271,272,273,274].

Transcription of Ty1 is regulated by several TFs (Gcr1, Ste12, Tec1, Mcm1, Tea1/Ibf1, Rap1, Gcn4, Mot3, and Tye7), shown in Table 2, and chromatin-remodeling complexes (Swi/Snf, SAGA, and ISWI) [244,246,247,275,276,277,278,279,280,281,282,283]. In haploid cells, transcription of Ty1 requires Ste12 and Tec1 that cooperatively bind to filamentation- and invasion-responsive element (FRE) sites located near the TATA box [242,243,275].

Furthermore, TF Mot3, a type of zinc finger protein, binds to the Ty912δ LTR. Ty912δ, a TE from Ty1, is integrated into the HIS4 promoter [284]. Mot3 regulates Ty912δ expression as either a repressor or activator [244]. Deletion of genes encoding the Gcn5 protein within the SAGA complex severely decreases Ty1 transcription [278]. Other Gcn genes also exert mild effects on Ty1 transcription [245,281,283,285]. Mcm1 interacts with the Tyl block II motif [246]. Generally, Mcm1 forms complexes with various TFs [286]. However, the Mcm1-binding motif is far from the binding motifs of other factors that interact with Mcm1, suggesting that Mcm1 may activate Ty1 transcription via an out-of-the-ordinary pathway [246]. The binding sites for the TF Rap1 are within the internal regulatory region of Ty1 elements. Rap1 plays an important regulatory role in the expression of Ty1 and Ty1-mediated adjacent genes. RAP1 strongly activates Ty1 transcription by forming a complex with Mcm1 [287]. The binding site of the TF Tea1 is located near the binding sites of Rap1 and Mcm1 [247], suggesting that the Tea1, Rap1, and Mcm1 complex regulates Ty1 transcription.

### 4.8. TFs Associated with the Drosophila (Gypsy)

In Drosophila, Ty3/gypsy is an LTR-containing TE on the X chromosome [288]. Motif 1 binding protein (M1BP) is a gypsy-activating TF ubiquitously expressed in *Drosophila* cells [248]. ChIP-seq revealed that M1BP interacts with the centrosomal protein (CP190) and activates motif 1-dependent transcription, followed by gypsy insulator activity [248].

## 5. Conclusions

The eukaryotic genome contains vestiges of ERVs, particularly at ~400,000 loci in humans [289]. Despite the loss of their innate virulence, ERVs possess full-length ORFs and can participate in cellular functions [4,115]. Indeed, ERVs are associated with cellular functions that may be beneficial or lead to disease progression in the host cells [58,59]. This raises the question of how they are activated in the genome. TFs are the prime elements that activate ERVs owing to the presence of the classical binding motif on their LTRs [94].

This review summarized the TFs that activate ERVs in various species and discussed their corresponding physiological roles. In cancer cells, Sox2, Oct4, and Nanog activate HERV-K 5Hs, leading to neurological illnesses. Also, c-Myc activates HERV-K, thereby facilitating conditions such as atypical teratoid rhabdoid tumors. Some TFs (such as YYI) possess both HERV activator and repressor activities. HERV-L and HERV-E, which are activated by the TFs DUX4 and HIF, respectively, are associated with cancer. On the other hand, DUX/DUX4 is a crucial regulator of ZGA gene expression and important for forming cultured two-cell-like cells. Alternatively, HERV-W and RLTR13D5 *env* protein (Syncytin-1) are known to participate in placental development in both humans and mice [183,231]. Human TFs, c-Myb, murine Cdx2, Eomes, and Elf5 are responsible for activating ERVs during placental formation. The Solo-LTRs are actively involved in cellular functions. For instance, upon IFN-γ stimulation, the solo-LTR12F switches on the full-length HERV-K102 in patients with cutaneous leishmaniasis [124], and IFN-γ and IFN-α together significantly activate HERV-K18 in T cells of patients with type-1 diabetes [126]. A similar phenomenon was observed for the upregulation of the HERV-H solo-LTR on chromosome 17q21 by Sp1, Oct4, Nanog, and LBP9. Prominent inducers of MERVs and MacERV6-LTR1 include Oct4, Sox2, and NANOG. In addition, Zscan4c, Gata4, Cdx2, and Elf5 activate MERVs. Notably, the most explored TF that triggers MacERV6-LTR1 is KLF4. The gypsy LTR in Drosophila is activated by M1BP and counteracted by the su protein. The zebrafish ERV EZR1 is primarily induced by Oct-1 and NF-κB. Moreover, yeast endogenous retrovirus (Ty1) is involved in cellular metabolism and is activated by Gcr1/4, Ste12, Tec1, Mcm1, Tea1/Ibf1, Rap1, Gcn1/4/5, Mot3, and Tye7 [275].

The fact that ERVs continue to be inscribed into the genome through the long process of biological evolution suggests that ERVs are not just fossils. In most animal species, ERVs are activated by TFs that are expressed early in development. Attractively, transcription factors expressed in early development, such as DUX, Sox2, and Oct4, are commonly used throughout the process of biological evolution. Considering that ERVs persist in utilizing these TFs during the long coexistence period of the host and virus confirms that ERVs may play a crucial role in early development. As the expression of these TFs is associated with various diseases, the causal relationship between ERVs and diseases needs to be carefully investigated in the future.

## Figures and Tables

**Figure 1 biomolecules-14-00280-f001:**
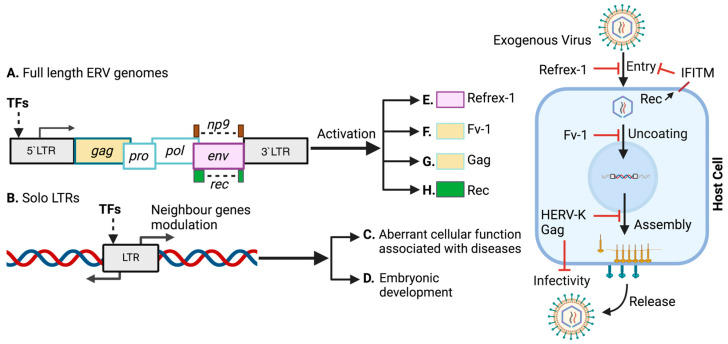
ERVs protect the host cells from the infection of exogenous retroviruses. (**A**) ERVs encode *gag*, *pro*, *pol*, *env*, and accessory genes *rec/np9*. (**B**) The solo LTRs lose the viral genes between 5′LTR and 3′LTR. (**C**) The solo LTRs modulate the genes associated with several diseases. (**D**) The solo LTRs modulate the genes associated with the placental and preimplantation developments. (**E**) The refrex-1 inhibits the viral entry in feline cells. (**F**) The Fv-1 inhibits the post-entry in murine cells. (**G**) The HERV-K Gag inhibits the viral assembly and viral infectivity. (**H**) The HERV-K Rev induces the IFITM expression to inhibit the viral infection. The graphic was created with BioRender.com. TFs: transcription factors; HERV-K: human endogenous retrovirus-K.

**Figure 2 biomolecules-14-00280-f002:**
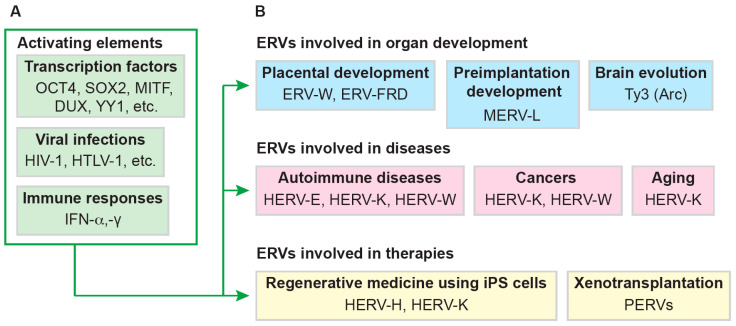
ERV-associated diseases, ERV evolution, and therapeutics. (**A**) ERVs transcription is activated by several factors such as transcription factors, viral infections, and immune responses. (**B**) The activation of ERVs is crucial for the placental development and preimplantation development. The ERVs expression is upregulated in several diseases. In addition, controlling the ERVs expression must be required for the regenerative medicine and xenotransplantation. MITF: microphthalmia-associated transcription factor; YY1: Yin Yang 1; HIV-1: human immunodeficiency virus type 1; HTLV-1: human T-lymphotropic virus type 1; IFN: interferon; ERV: endogenous retrovirus; HERV: human endogenous retrovirus; iPSCs: induced pluripotent stem cells; PERV: porcine endogenous retrovirus.

**Table 1 biomolecules-14-00280-t001:** List of human endogenous retroviruses, their transcription factors, chromosomal location, and associated functions. HERV: human endogenous retrovirus; ERV: endogenous retrovirus; DUX: double homeobox 4; HIF: hypoxia-inducible transcription factor; NF-κB: nuclear factor-κB; YY1; Yin Yang 1; chr: chromosome; AT/RT: atypical teratoid rhabdoid tumor; ALS: amyotrophic lateral sclerosis; IFN: interferon; SLE: systemic lupus erythematosus.

Species	Viruses	Transcription Factors	Chromosome	Disease/Function	Reference
**Human**	HERV-K5Hs, 5A, 5B	Sox2, Oct4, and Nanog	chr5q33.3	Neurodegenerative disease	[107,108]
	HERV-K and HERV-L	DUX4	chr17q and chr7p22	FSHD	[115]
	HERV-K5Hs, and HERV-1	HIF		Kidney cancer	[116]
	HERV-K5Hs	c-Myc	chr7p22.1a and chr7p22.1b	AT/RT	[95]
	HERV-K5Hs	NF-κB and IRF1		ALS	[117]
	HERV-K5Hs	PR/Estradiol		Breast cancer	[118]
	HERV-K5Hs and HERV-E	Sp1 and Sp3		Skin cancer	[103]
	HERV-K5Hs	TDP-43		ALS	[119]
	HERV-K5Hs, 5A, 5B	p53, p60, and p65		Leukemia, ovarian, and colorectal cancers	[101,104]
	HERV-K5Hs	YY1		Cancers	[120,121]
	HERV-K	NF-AT		HIV associated malignancy	[122]
	HERV-K	MITF-M	chr1	Melanoma	[123]
	HERV-K	STAT-1 and IRF1		Inflammatory disease and IFN-γ signaling	[124]
	HERV-K	USF-1		Wound repair regulation	[108]
	HERV-K	Tax		Associated with opportunistic infection	[125]
	HERV-K	Tat		Associated with opportunistic infection	[122]
	HERV-K18	IFN-α	chr1q21.2-1q22	Type-1 diabetes	[126]
	HERV-K102	IFN- γ		Leishmaniasis	[124]
	HERV-L	HNF-1		Colon cancer	[79]
	HERV-W	c-Myb/HOXA5	chr7q21.2	Tumor progression	[127]
	HERV-W	OCT-1		Cell abnormalities	[128]
	HERV-W	Sp1 and Sp3		Lung fibroblasts	[129]
	HERV-W	GCM-a	chr7q21-7q22	Placental formation	[130]
	HERV-E	HIF-2α HIFs, HIF-1α, HIF-2α, and HIF-1β		Kidney cancer	[131]
	HERV-E	*NAFT1*	chr19p12	SLE	[132]
	HERV-H	Sp1, GC box, and TATA box		Breast cancer	[133,134,135,136]
	HERV-H	LBP9, Oct4, Nanog, and Klf4		Chromosome duplication	[137]
	HERV-T	GATA4, and FOXA2		Endoderm specification	[138]
	ERV-9	GATA-2, NF-Y, and MZF1		Chromatin remodeling	[139]

**Table 2 biomolecules-14-00280-t002:** List of species-specific endogenous retroviruses excluding humans, their transcription factors, chromosomal locations, and the associated functions. MERV: murine endogenous retrovirus; LTR: long terminal repeat; PERV: porcine endogenous retrovirus; SERV: simian endogenous retrovirus; EZR: expressed-zebrafish-retroelement group type 1; ZFERV: zebrafish endogenous retrovirus; TRIM28: tripartite motif-containing protein 28; NF-κB: nuclear factor-κB; chr: chromosome.

Species	Viruses	Transcription Factors	Chromosome	Disease/Function	Reference
**Mouse**	MERV-L LTR1	Zscan4c		Impair the embryonic development	[216]
	MERV-L LTR	DUX			[140]
	MERV-L RLTR9B2, RLTR9D, and RLTR9E	Esrrb, Klf4, and Sox2		Abnormality in cell pluripotency	[230]
	MERV-L LTR1	KDM1A		Impair the embryonic development	[225]
	MERV-RLTR13D5	Cdx2, Eomes, and Elf5		Regulate the placental development	[231]
**Pig**	PERV-A, PERV-B, and PERV-C LTR	NF-AT, Oct-1, Ets-1, v-myb, HFH-3, NF-1, AP-1, NF-Y, and AP-1C		Impair the embryonic development	[232,233]
**Monkey**	Cer-SERV-1 LTRs	AREB6, Brn-2, CDP, CREB, E2F, FoxD3, and FoxJ2		Embryonic development	[234]
	Cer-SERV-2 LTRs	AREB6, COMP1, CREB, NF-1, RFX1, Pax6, and v-Myb		Embryonic development	[235]
	MacERV6-LTR1	ESRRB		Embryonic development	[236]
**Bovine**	BoERV3, BoERV9, and BoERV16	Rxra	chr2 and chr4	Endocrine function	[237]
	MER41_BT	Oct4, NFY, Klf4, OTX2, TEAD, and STAT1			[238]
	Bov-A2	POLR2A			[239]
	BoRV CH15	NF-1		Cattle encephalitis	[240]
**Zebra Fish**	EZR1	NF-κB	chr4 and chr5	Cardiac malfunction	[241]
	EZR1	TCF-11, Nkx-2.5, GATA-1, MZF-1, and Ikaros-2		Lymphocyte maturation	[241]
**Yeast**	Ty1 LTR	Ste12 and Tec1		Cell cycle regulation	[242,243]
	Ty912d LTR	Mot3		Gene activation or repression	[244]
	Ty1 LTR	Gcn4, and Gcr1		Glucose metabolism	[245]
	Ty1 LTR	Mcm1		Cellular metabolism	[246]
	Ty1 LTR	Tea1 and Rap1		Cellular metabolism	[247]
**Drosophila**	Ty3/Gypsy	M1BP	chrX		[248]

## Data Availability

No new data were created.

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
