# Peer review of "Species-Specific Transcription Factors Associated with Long Terminal Repeat Promoters of Endogenous Retroviruses: A Comprehensive Review"

_biomolecules, 2024, doi:10.3390/biom14030280_

Round 1
Reviewer 1 Report
Comments and Suggestions for Authors
In this manuscript, Md Jakir Hossain et al. describe TFs associated with long terminal repeat promoters of endogenous retrovirus. The theme of the Review is interesting. The manuscript discusses TFs actively associated with ERVs, in multiple systems of study. The authors provide mechanistic descriptions regarding the binding of many TFs on ERVs and their connection with critical phenomena, including those that characterize human diseases. Overall, this is an interesting Review.
The manuscript has to be improved in some points outlined below:
Major Points
[1] Introduction Section: Page 2, 4th Paragraph.
“As demonstrated by earlier studies, TFs or sequence-specific DNA-binding factors
are molecules that bind to a specific DNA sequence and modulate the transcription of DNA to mRNA (90, 91).”
The authors have to include a better introductory sentence for TFs. In addition, they have to highlight further the impact of TFs on gene expression regulation.
[2] Figure 1: The resolution of the Figure has to be improved. In addition, a more detailed Figure Legend is required.
[3] Table 1 and Table 2: The resolution of the Tables has to be improved.
[4] Results Section: Paragraph 2.2
“Overall, it can be said that both NF-κB and NF-AT activation in response to HIV-1 Tat drive transcription from the HERV-K promoter in HIV-1-infected patients (140).”
The authors can discuss more the part of the manuscript that describes HIV activation.
Minor Points
[1] The authors can discuss more about the evolutionary conservation of those TFs.
[2] The authors can include a graphical representation of the phenomena that are described in the manuscript.
Comments on the Quality of English Language
The authors have to correct a spelling error. Results Section 2.1; First Paragraph: “These finding suggest”
Author Response
Major Points
[1] Introduction Section: Page 2, 4th Paragraph.
“As demonstrated by earlier studies, TFs or sequence-specific DNA-binding factors are molecules that bind to a specific DNA sequence and modulate the transcription of DNA to mRNA (90, 91).” The authors have to include a better introductory sentence for TFs. In addition, they have to highlight further the impact of TFs on gene expression regulation.
- Thank you for your great suggestion. We modified and highlight the inpact of TFs in the paragraph (Page 3, line 934-944). I hope that this modification answers the comments intended by reviewer 1.
[2] Figure 1: The resolution of the Figure has to be improved. In addition, a more detailed Figure Legend is required.
- We changed the resolution of Figures and added the further information in figure legend.
[3] Table 1 and Table 2: The resolution of the Tables has to be improved.
- We changed the resolutions of Table 1 and Table 2.
[4] Results Section: Paragraph 2.2
“Overall, it can be said that both NF-κB and NF-AT activation in response to HIV-1 Tat drive transcription from the HERV-K promoter in HIV-1-infected patients (140).” The authors can discuss more the part of the manuscript that describes HIV activation.
- Thank you for your great assists. We described the further information (Page 7-8, line 3170-3719).
Minor Points
[1] The authors can discuss more about the evolutionary conservation of those TFs.
- We described the further information (Page 15, line 7636-7640).
[2] The authors can include a graphical representation of the phenomena that are described in the manuscript.
- We added the graphical representation in Figure 1.
Reviewer 2 Report
Comments and Suggestions for Authors
It is an interesting review, regarding the participation of the expression of endogenous retroviruses related to the LTR region and its association with the presentation of diseases. I consider that it may be suitable for publication in the journal Biomolecules.
I suggest the authors carefully review the writing of the article. Due to the absence of line numbering, it is not easy to indicate the paragraphs where I suggest improving the writing or correcting errors, for example.
The sources of these somatic expressions are the leaky or ERV LTRs (58). Forbye, the induction of HERV-K expression by the viral infection……..
The LTRs of the human-specific HERV-K (HML-2) LTR have the essential features………
The results based on the activation/silencing of HERV-K LTR5Hs system using the CARGO with…………….
Since the human PERV-A receptor 1 and 2 (huPAR1 and huPAR1, respectively)….
I would like the authors to consider including information on endogenous feline retroviruses, which although there is not as much information as on other species, there are modest contributions on the LTR region and the possibility of participation in the development of disease and especially in the generation of new viral populations that can cause significant disease in cats. I list some publications about it.
Berry et al., B.T. Berry, A.K. Ghosh, D.V. Kumar, D.A. Spodick, P. Roy-Burman
Structure and function of endogenous feline leukemia virus long terminal repeats and adjoining regions J. Virol., 62 (1988), pp. 3631-3641.
Tandon R, Cattori V, Pepin AC et al (2008) Association between endogenous feline leukemia virus loads and exogenous feline leukemia virus infection in domestic cats. Virus Res 135:136–143.
Acevedo‑Jiménez G., Sarmiento‑Silva R., Alonso‑Morales R., Córdova‑Ponce R., Ramírez‑Álvarez H. Detection and genetic characterization of feline retroviruses
in domestic cats with different clinical signs and hematological alterations. Archives of Virology (2023) 168:2.
Author Response
I suggest the authors carefully review the writing of the article. Due to the absence of line numbering, it is not easy to indicate the paragraphs where I suggest improving the writing or correcting errors, for example.
- Thank you for your comments. We modified all your suggested sentences. And we proceeded the proofreading by editage company to find all writing errors.
I would like the authors to consider including information on endogenous feline retroviruses, which although there is not as much information as on other species, there are modest contributions on the LTR region and the possibility of participation in the development of disease and especially in the generation of new viral populations that can cause significant disease in cats. I list some publications about it.
- Thank you for your suggestion. We added your suggested citations (ref#253, 254, 255) and described in feline section 4.5 (Page 14, line 7043-7060).
Berry et al., B.T. Berry, A.K. Ghosh, D.V. Kumar, D.A. Spodick, P. Roy-Burman. Structure and function of endogenous feline leukemia virus long terminal repeats and adjoining regions J. Virol., 62 (1988), pp. 3631-3641.
Tandon R, Cattori V, Pepin AC et al (2008) Association between endogenous feline leukemia virus loads and exogenous feline leukemia virus infection in domestic cats. Virus Res 135:136–143.
Acevedo‑Jiménez G., Sarmiento‑Silva R., Alonso‑Morales R., Córdova‑Ponce R., Ramírez‑Álvarez H. Detection and genetic characterization of feline retroviruses in domestic cats with different clinical signs and hematological alterations. Archives of Virology (2023) 168:2.
Reviewer 3 Report
Comments and Suggestions for Authors
This review summarized the transcription factors recognized the LTR region of ERVs. It is timely review of current findings. However, there are a few points that should be addressed before publication.
1. It is important to discuss the role of Dux as an activator of MERVL in mice. Dux directly binds to MERVL LTR and indirectly regulates MERVL by acting downstream of p53 or Pim3. Therefore, the authors also should discuss the function of Dux in mouse.
2. In the section titled "ERV LTR is required for the transcription of neighboring genes," the authors primarily focus on human ERVs. However, it is worth mentioning that the LTR of mouse ERVs, such as MERVL, has also been identified as a promoter driving cryptic transcription of genes. The authors should discuss about this.
3. I suggest revising the title of Table 2 to clarify that it includes information from other species, excluding humans.
4. The first line of Table 2 is missing the species information.
5. The authors should verify whether Trim28 functions as a transcription factor or a co-repressor and make the necessary clarification in the manuscript.
Author Response
This review summarized the transcription factors recognized the LTR region of ERVs. It is timely review of current findings. However, there are a few points that should be addressed before publication.
- It is important to discuss the role of Dux as an activator of MERVL in mice. Dux directly binds to MERVL LTR and indirectly regulates MERVL by acting downstream of p53 or Pim3. Therefore, the authors also should discuss the function of Dux in mouse.
- Thank you for your suggestion. We discuss the role of Dux, p53 and Pim3 as MERV-L (Page 11, line 5454-5459). The function of Dux is described in conclusions (Page 15, line 7568-7570).
- In the section titled "ERV LTR is required for the transcription of neighboring genes," the authors primarily focus on human ERVs. However, it is worth mentioning that the LTR of mouse ERVs, such as MERVL, has also been identified as a promoter driving cryptic transcription of genes. The authors should discuss about this.
- Thank you for your suggestion. We discuss the importance of LTR in mouse in section 4 (Page 10-11, line 4806-5429).
- I suggest revising the title of Table 2 to clarify that it includes information from other species, excluding humans.
- We modified the tile of Table 2.
- The first line of Table 2 is missing the species information.
- We modified the line in Table 2.
- The authors should verify whether Trim28 functions as a transcription factor or a co-repressor and make the necessary clarification in the manuscript.
- We notice the critical mistake in conclusion. We modified the mistake (Page 15, line 7569-7570). And we described the Trim28 functions in detailed (Page 11, 5437-5445).
Round 2
Reviewer 1 Report
Comments and Suggestions for Authors
The authors have addressed my comments and suggestions.